# The Lyme disease bacterium, *Borrelia burgdorferi*, stimulates an inflammatory response in human choroid plexus epithelial cells

Derick Thompson, Jordyn Sorenson, Jacob Greenmyer, Catherine A. Brissette, John A. Watt ◑ *

Department of Biomedical Sciences, University of North Dakota, Grand Forks, North Dakota, United States of America

* john.watt@und.edu

**Data Availability Statement:** If the data are held or will be held in a public repository, include URLs, accession numbers or DOIs. If this information will

## Abstract

The main functions of the choroid plexus (CP) are the production of cerebral spinal fluid (CSF), the formation of the blood-CSF barrier, and regulation of immune response. This barrier allows for the exchange of specific nutrients, waste, and peripheral immune cells between the blood stream and CSF. *Borrelia burgdorferi* (*Bb*), the causative bacteria of Lyme disease, is associated with neurological complications including meningitis–indeed, *Bb* has been isolated from the CSF of patients. While it is accepted that *B. burgdorferi* can enter the central nervous system (CNS) of patients, it is unknown how the bacteria crosses this barrier and how the pathogenesis of the disease leads to the observed symptoms in patients. We hypothesize that during infection *Borrelia burgdorferi* will induce an immune response conducive to the chemotaxis of immune cells and subsequently lead to a pro-inflammatory state with the CNS parenchyma. Primary human choroid plexus epithelial cells were grown in culture and infected with *B. burgdorferi* strain B31 MI-16 for 48 hours. RNA was isolated and used for RNA sequencing and RT-qPCR validation. Secreted proteins in the supernatant were analyzed via ELISA. Transcriptome analysis based on RNA sequencing determined a total of 160 upregulated genes and 98 downregulated genes. Pathway and biological process analysis determined a significant upregulation in immune and inflammatory genes specifically in chemokine and interferon related pathways. Further analysis revealed downregulation in genes related to cell to cell junctions including tight and adherens junctions. These results were validated via RT-qPCR. Protein analysis of secreted factors showed an increase in inflammatory chemokines, corresponding to our transcriptome analysis. These data further demonstrate the role of the CP in the modulation of the immune response in a disease state and give insight into the mechanisms by which *Borrelia burgdorferi* may disseminate into, and act upon, the CNS. Future experiments aim to detail the impact of *B. burgdorferi* on the blood-CSF-barrier (BCSFB) integrity and inflammatory response within animal models.

only be available after acceptance, indicate this by ticking the box below. For example: All gene files are available from the GEO database.

**Funding:** C.A.B. was supported by National Institutes of Health Cobre grant number P20GM113123. J.A.W was supported by Natioanal institutes of Health Cobre grant number P20GM104360. This work was also supported by the National Institute of General medical Sciences grant U54GM128729 and National Institutes of Health grant 2P20GM104360-06A1. The funders had no role in study design, data collection and analysis, decision to publish, or preparation of the manuscript.

**Competing interests:** The authors have declared that no competing interests exist.

## Introduction

Lyme disease, caused by the spirochete *Borrelia burgdorferi* (*Bb*), is the most commonly reported vector-borne disease in the United States–with 30,000 cases being reported to the CDC annually [1, 2]. Like many other diseases, *B. burgdorferi* is subject to underreporting, as demonstrated by two studies conducted by the CDC that concluded an estimated 300,000 individuals are infected with Lyme disease each year in the U.S [3, 4]. Associated medical costs of initial treatment and extended healthcare for ongoing symptoms attributed to post-treatment Lyme disease syndrome (PTLDS) are estimated to be between $712 million—$1.3 billion each year [5].

The symptoms of Lyme disease can range from erythema migrans to more systematic disorders such as arthritis and neurological complications, termed neuroborreliosis [6, 7]. Manifestations of neuroborreliosis include radiculoneuritis, meningitis, and facial palsy [8–10]. It is well-documented that *B. burgdorferi* is capable of penetrating into the central nervous system (CNS). This is evident from the direct detection of the pathogen within the cerebral spinal fluid, usually performed by lumbar puncture, followed by bacterial culture or PCR [11]. Furthermore, other methods suggest of CNS invasion—the detection of intrathecal antibodies, an increase in peripheral immune cells, such as lymphocytic pleocytosis, and the presence of the chemoattractant *cxcl13* [12–15]. Though methods of detection and diagnosis of neuroborreliosis continue to grow, very little is known about the mechanisms by which *B. burgdorferi* enters the CNS and the pathophysiology of the disease. *B. burgdorferi* does not produce or secrete any known toxins and it is suggested that the host inflammatory response elicited by the bacteria is a factor in the pathogenesis of the disease [16–18]. Explants and primary cultures of dorsal root ganglia tissue from rhesus macaques that were incubated with *Borrelia burgdorferi* showed an increase in inflammatory cytokines *ccl2*, *il-6*, and *il-8*, as well as the apoptosis of sensory neurons [19]. The correlation between inflammation and the pathology of the disease is also observed in the inflammation and subsequent apoptosis of oligodendrocyte cultures following *Bb* infection [20]. This is further seen in the cerebral spinal fluid (CSF) of patients with confirmed neuroborreliosis that show increases in chemokines such as *ccl2*, *ccl5*, and *cxcl1* [21–23]. The presence of these chemokines may indicate a role for these factors in the host immune response, notably immune cell trafficking.

The choroid plexus (CP) is one such complex that has been implicated in the trafficking of immune cells across its blood-CSF-barrier (BCSFB). In addition to its role in the formation of the BCSFB, it is the major producer of CSF [24, 25]. The CP is a highly vascularized structure within the ventricles of the brain, and unlike the blood-brain barrier (BBB), the capillaries within the choroid plexus are highly fenestrated. Instead, the epithelial layer is responsible for the selective permeability of the BCSFB through the formation of tight and adherens junctions [26]. An interesting characteristic of the choroid plexus is the presence of immune cells on the basolateral side within the stromal matrix–this includes dendritic cells and macrophages (Fig 1) [27–30]. Further illustration of the immune-surveillance role of the choroid plexus is shown in the presence of cell adhesion molecules on CP epithelium and not the neighboring endothelium, which mediate the binding of immune cells [31]. The transmigration of macrophages and peripheral blood mononuclear cells (PBMCs) across the choroid plexus epithelium was observed in transwell and explant cultures in the presence of feline immunodeficiency virus [32]. In a human barrier model of the choroid plexus, the transepithelial migration of polymorphonuclear neutrophils and monocytes were observed following bacterial infection (*Neisseria meningitidis*) [32]. It is understood that the context of a bacterial infection and the inflammatory profile of the CSF determines the severity and overall outcome of patients [33–35]; therefore, due to the choroid plexus's anatomical position in separating the periphery and

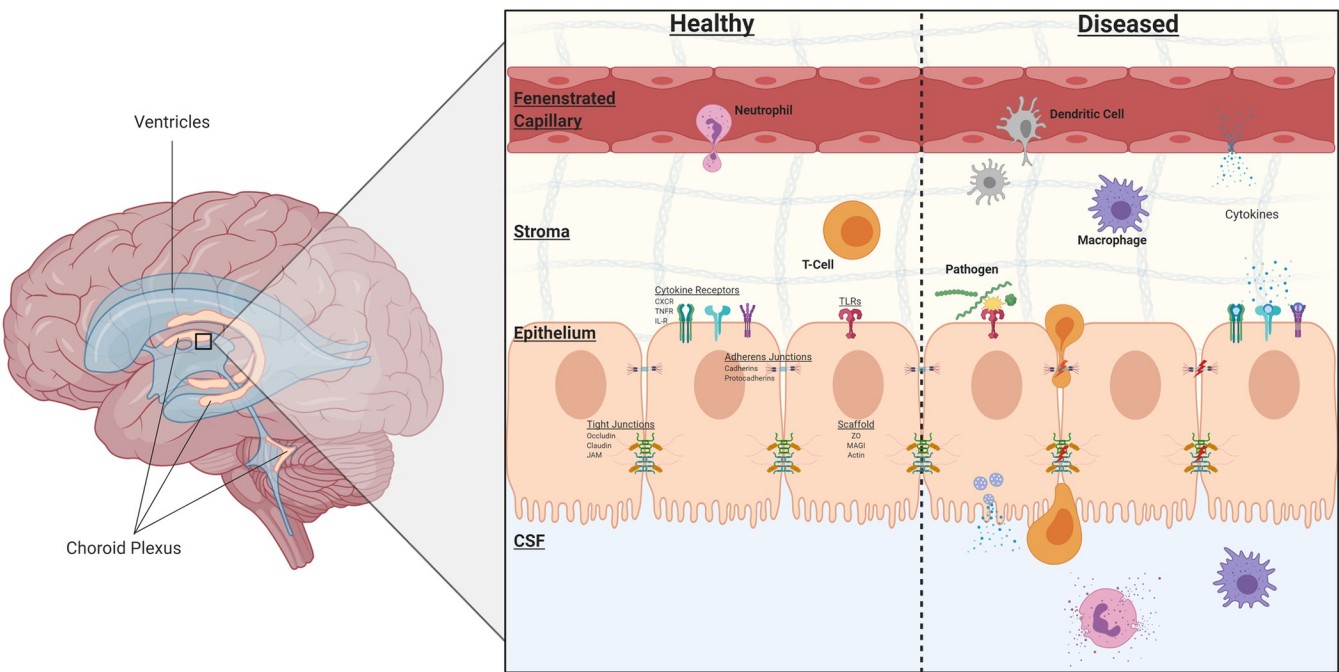

**Fig 1. Structural features of the choroid plexus.** Under healthy conditions, the choroid plexus maintains peripheral immune cells within the stromal matrix as a form of immunosurveillance. Multiple receptors are present on the epithelium, including cytokine and pattern recognition receptors. The BCSFB is formed by the epithelium through tight junctions, adherens junctions, and desmosomes. During infection or disease states, these junctions may be altered, immune cells may transmigrate across the epithelium, and cytokines from both the epithelial cells and activated immune cells are released. Many of these cytokines induce a pro-inflammatory response and act as chemoattractants for innate and adaptive immune cells. Created with Biorender.com.

CNS, as well as its secretory, barrier, and immune response roles, the intent of this project was to investigate the effects of *Borrelia burgdorferi* infection on choroid plexus epithelial cells.

The overall aim of this study is to determine the transcriptome profile of primary human choroid plexus epithelial cells (HCPECs) during *B. burgdorferi* infection. Using HCPECs in culture, we demonstrated differential expression of 258 genes following infection with *B. burgdorferi* after 48 hours. Functional and pathway analysis of these transcriptional changes revealed upregulation of host inflammatory and immune responses, related to interferon, chemokine, and cytokine pathways, as well as immune cell trafficking and activation. Although this model does not provide a barrier context, interestingly, functional and cellular components involving cell-cell junctions and tight junctions were seen to be downregulated. Here we present our findings of differential gene expression in HCPECs following infection with *B. burgdorferi*.

## Methods

### Bacteria culture

The *Borrelia burgdorferi* strain B31-MI-16 is an infectious clone which was previously sequenced and described [36, 37]. Bacteria cultures were grown to approximately 1 x $10^7$ bacteria/ml in modified Barbour-Stoenner-Kelly (BSK-II) medium supplemented with 6% rabbit serum at 34˚C and used at passage 2.

### Cell culture

HCPECs were obtained from ScienCell Research Laboratories (Carlsbad, CA; catalog #1310). Commercially available human primary cell cultures and protocols used throughout this study

followed the University of North Dakota IRB guidelines outlined in form 504 "Categories of Research", section 2.19 "Commercially Available Human Biological Specimens (45 CFR 46.102, 46.103, and 46.116" and therefore do not require IRB review. Purity of cells was assessed and confirmed by immunofluorescence, the following primary antibodies and concentrations were used: Rabbit anti-Prealbumin (TTR) conjugated to Alexa Fluor 488 (Abcam, catalog #ab199074; Conc. 1:50), Mouse anti-α-Tubulin (Sigma, catalog #T61999; Conc. 2 μg/ml), Mouse anti-CK18 (Abcam, catalog #ab82254; Conc. 1:50). The secondary antibody, Donkey anti-Mouse conjugated to Alexa Fluor 594 (Jackson Immunoresearch, catalog #715-585-151, Conc. 1:100), was used for anti-CK18 and anti-α-Tubulin antibodies. For immunofluorescence, cells were grown on glass coverslips and fixed with a 4% paraformaldehyde solution for 15 minutes and stored in PBS at 4˚C. Permeabilization was performed with a 0.1% Triton-PBS solution for 10 minutes and blocked with a 10% Donkey (Jackson immunoresearch, catalog #017-000-121) or Goat Serum (Jackson Immunoresearch, catalog #005-000-121) in 0.1% Tween-PBS solution for 1 hour. Primary antibodies were incubated overnight at 4˚C. Secondary antibodies were incubated for 1 hour at room temperature. DAPI Fluoromount-G (SouthernBiotech, catalog #0100–20) was used for nuclei staining. Western blot analysis was performed on control and infected cell lysates to determine presence of TTR and CK18 –previously mentioned primary antibodies were used in conjunction with the secondary antibodies: Donkey anti-rabbit conjugated to peroxidase (Jackson Immunoresearch, catalog #711-035-152, Conc. 1:200,000) and Goat anti-mouse conjugated to peroxidase (Jackson Immunoresearch, catalog #115-035-003, Conc. 1:200,000). Signal was produced by SuperSignal West Femto Maximum Sensitivity Substrate kit from ThermoFisher Scientific (catalog #34094) and imaged on a Licor Odyssey Fc Imaging System.

Cells were maintained in tissue-treated vented cap T-75 flasks (Corning, catalog #430641U) in epithelial cell medium (ScienCell, catalog #4101), containing antibiotics penicillin (100 units/ml) and streptomycin (100 ug/ml) (ScienCell, catalog #0503), 2% fetal bovine serum (ScienCell, catalog #0010), and EpiCGS (ScienCell, catalog #4125). Two groups were used for these experiments–Control, non-infected (n = 3) and Infected, 48 hours (n = 3); a total of 6 samples. Cells were incubated at 37˚C and used at passage 3 at approximately 80% confluence for stimulation by *Bb*. Prior to infection, the cell cultures were washed 3 times with sterile Dulbecco's phosphate buffered saline (DPBS) and the medium replaced with antibiotic-free epithelial cell medium. Cell cultures were stimulated with *B. burgdorferi* at a multiplicity of infection (MOI) of 10:1 (bacteria:cells) for 48 hours. Mean cell count of HCPECs was determined using an automated cell counter (Life Countess II, catalog #AMQAX1000) to determine appropriate number of bacteria for infection. Control non-treated flasks were prepared identically without infection. Light microscopy was used to monitor cell morphology and confluency. Additionally, cell proliferation was monitored using an MTS assay (Abcam, catalog #ab197010) and colorimetric plate reader in a 96-well plate under identical conditions. Apoptosis and necrosis were determined by fluorescence microscopy using Apopxin Green Indicator (Apoptosis) and 7-AAD (Necrosis) on cells grown in 24-well plates on glass coverslips under identical conditions (Abcam, catalog #ab176749) and manually counted.

## RNA isolation

RNA was isolated via phenol-chloroform extraction and using the RNeasy Mini kit from Qiagen (catalog #74106) according to the manufacturer's instructions. In short, cell medium was removed from cultures and used for later protein analysis; 1 ml of trizol was added directly to each flask and a cell scraper was used to fully lyse all cells. Homogenized cells were transferred

to RNase-DNase free 1.5 ml Eppendorf tubes, where chloroform was added for 2 minutes and centrifuged at 12,000 x g for 15 minutes at 4°C. The upper aqueous phase was mixed with 70% ethanol and placed in a Qiagen RNeasy Mini column. The flow-through was discarded and the bound RNA fraction remaining on the column membrane was further washed and processed per Qiagen's instructions. Genomic DNA was removed with DNA digestion with RNase-free DNase Set (Qiagen, catalog #79254). RNA quality was assessed with a NanoDrop and integrity assessed by gel electrophoresis on a 2% agarose gel.

## Library construction and RNA sequencing

Isolated total RNA, as described above, underwent further quality control and purification to obtain mRNA. To assess RNA integrity of total RNA, samples were placed in an Agilent 2100 Bioanalyzer with the RNA 6000 Nano kit (catalog #5067–1511)–all samples passed with an RNA integrity number (RIN) of $\geq 8.9$. To obtain a more accurate concentration, total RNA samples were run on a broad range Qubit 2.0 Fluorometer. mRNA was enriched from total RNA samples using the NEBNext Poly(A) mRNA Magnetic Isolation Module (catalog #E7490S)–in short, oligo d(T) beads are used to bind the poly(A) tail of eukaryotic mRNA. The NEBNext Ultra II RNA-seq library kit (catalog #E7775S) was used for library construction. Libraries were checked for quality and adaptor contamination on the bioanalyzer with the Agilent DNA 1000 kit (catalog #5067–1504). Library concentration was assessed with a BioTek Gen5 Wellplate reader with the Quant-iT PicoGreen dsDNA Assay kit (catalog #P11496). All samples, 3 control and 3 infected, were then pooled and sent to Novogene (https://en.novogene.com) for sequencing. An Illumina HiSeq 4000 was used for 150 bp paired-end sequencing.

## RNA data analysis

Raw fastq files were received from Novogene and initial quality control was assessed using FastQC version 0.11.2 [38]. All samples passed initial QC following adaptor trimming using Trimmomatic [39]. Reads were aligned to the human (hg19) assembly using Hisat2, version 2.1.0 [40] and indexed by Samtools, version 1.9 [41]. Differential gene expression analysis was performed using DESeq2, version 1.24.0 [42], with an FDR of 0.05 or lower, and no fold change cut-off. Network mapping and functional analysis was performed with STRING database, version 11.0 [43] and verified with PANTHER, version 14.1 [44]. STRING utilizes Gene Ontology [45, 46] to determine functional enrichments within our networks.

Pathway analysis was performed using Signaling Pathway Impact Analysis (SPIA), version 2.36.0 [47, 48], which brings fold change and gene function into context. SPIA uses the Kyoto Encyclopedia of Genes and Genomes [49] (KEGG) database to determine impact of DEGs on the respective pathway based on gene enrichment and topology of the pathway. Pathway enrichment is determined from the total number of genes within a specific pathway compared to the Number of Differential Expressed genes (NDE) observed within that pathway; significance of pathway enrichment was set at pNDE < 0.05. Furthermore, the topology of a pathway is taken into consideration to determine the impact of DEGs within that pathway. The perturbations (PERT) of a pathway caused by DEGs is determined based on the location of these genes within the pathway; significance was set to pPERT < 0.05. Overall global significance (pG) was determined from pNDE and pPERT. Two forms of statistical corrections pG were performed–a Bonferroni correction (pGFWER) and a false discovery rate (FDR) correction (pGFDR). To determine significance of a pathway, pGFDR < 0.05 was considered.

## Validation of RNA-seq using RT-qPCR and cDNA synthesis

Selected individual transcripts were confirmed using PCR primer sets (Qiagen, catalog #330001). cDNA from RNA samples were synthesized using Qiagen's First Strand Kit (catalog #330404). Each reaction was performed following the RT$^2$ qPCR Primer Assay instructions– each reaction contained 1 μl of the primer mix at 10 μM for each gene of interest, 12.5 μl RT$^2$ SYBR Green Mastermix, 1 μl cDNA, and 10.5 μl Nuclease-free water, for a total reaction volume of 25 μl. qPCR was initiated with a single 10 min cycle at 95˚C for initial denaturing and activation of polymerase. Following this, 40 cycles of 15 seconds at 95˚C and 1 minute at 60˚C was performed and fluorescent data was collected at the end of each cycle. Melt curve analysis was performed at the end of the reaction using the following conditions: 95˚C, 1 min; 65˚C, 2 mins; 65˚C to 95˚C step-wise at 2˚C/min. Expression levels of transcripts were compared and normalized to the housekeeping gene *actb* (β-actin). Relative gene expression between treated and untreated sample groups were compared using the $2 ^{–\Delta\Delta CT}$ method. All samples were analyzed in triplicate from three biological replicates.

## Supernatant protein analysis by enzyme-linked immunosorbent assays

Supernatants from cultures were removed as previously mentioned following treatment. Samples were aliquoted and stored at -20˚C until use. ELISAs were performed following manufacturer's instructions (R&D Systems, DuoSet ELISA). Briefly, plates were coated and incubated overnight at room temperature with 100μl of capture antibody. Following aspiration of this antibody and washing, 100 μl of standards and sample were added to each well. Plates were than incubated at room temperature for 2 hours, aspirated, and then washed. 100 μl of conjugated detection antibody was then added to each well and incubated for 2 hours at room temperature. Colorimetric detection was performed after the addition of a chromogenic substrate and stop solution. Plates were read at a wavelength of 450 nm on a BioTek Epoch plate reader.

## Statistical analysis

Differential gene expression of RNA sequencing data was determined by DESeq2. Briefly, DESeq2 utilizes an empirical Bayes approach and makes the assumption that genes of similar transcript levels will show similar variability. Through this, the package can control for replicate variability at each specific gene by taking into consideration the average variability of similarly expressed genes, and account for sample size. Furthermore, DESeq2 performs a Benjamini-Hochberg adjustment resulting in an adjusted p-value (padj) also called a false discovery rate (FDR). Genes with an FDR < 0.05 were considered significant.

Statistical analysis between control and infected groups for both RT-qPCR and ELISA was performed using an unpaired student's t-test using GraphPad Prism Version 8. Additional post-hoc analysis was performed on RT-qPCR to correct for multiple t-test comparisons. The method of Benjamini, Krieger, and Yekutieli was used to determine the FDR–this method is an updated version of the previously mentioned Benjamini-Hochberg adjustment [50]. Transcripts for RT-qPCR were considered significant if FDR < 0.05 (*) Protein levels from ELISA were considered significant if p < 0.05 (*).

# Results

Prior to infection, the identity and purity of HCPECs was assessed and confirmed through immunohistochemistry (IHC) with cytokeratin 18 (CK18), an epithelial cell marker, and transthyretin (TTR), a marker for choroid plexus epithelial cells and specific hepatocytes (Fig 2A and 2B)– 98% of cells were CK18 and TTR positive [51]. Fig 2A indicates the colocalization of

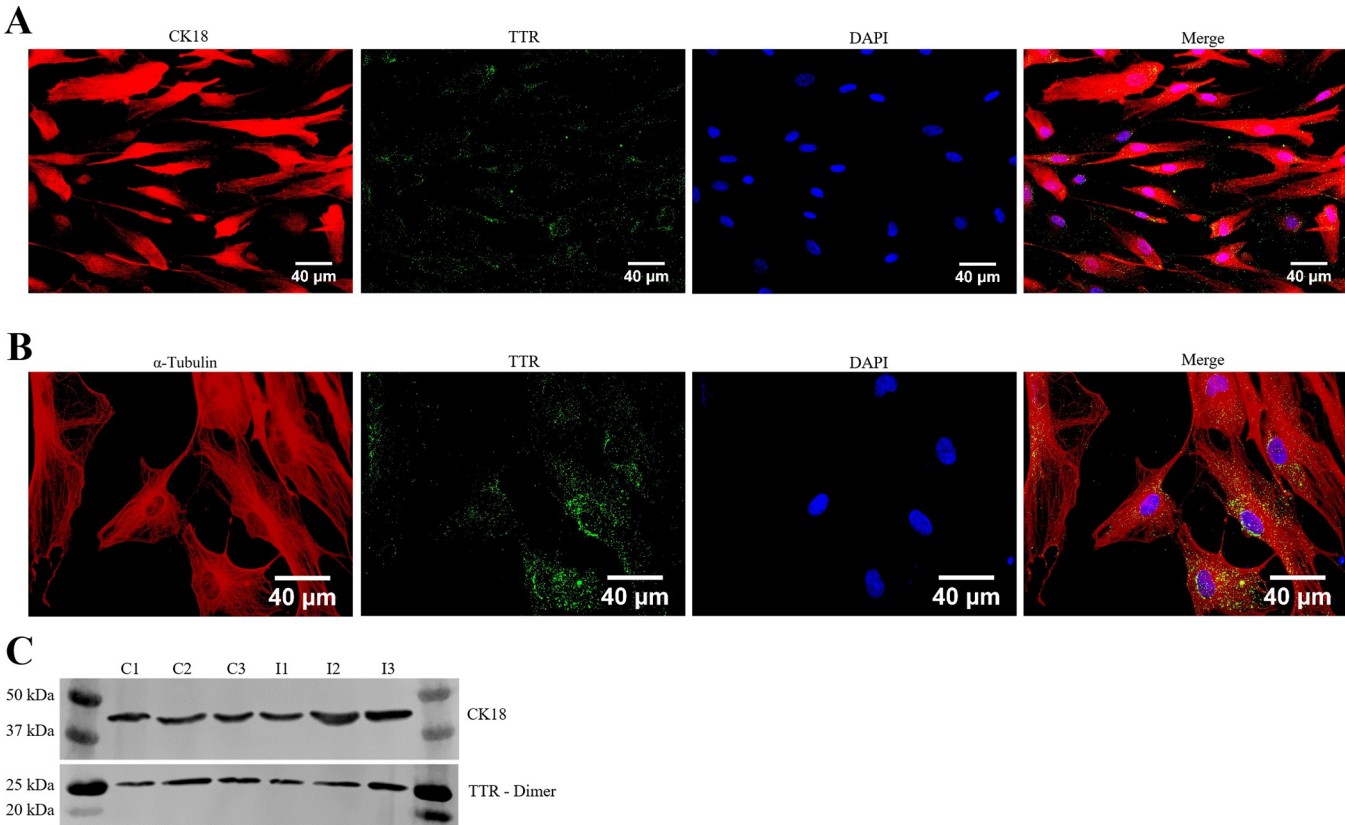

**Fig 2. Characterization of primary HCPECs.** (A) HCPECs were identified by immunostaining with cytokeratin 18 (CK18—red), an epithelial cell marker, and transthyretin (TTR–green), a transport protein predominantly expressed by choroid plexus epithelial cells and hepatocytes. Nuclei were stained with DAPI (blue). Magnification: 40x, Scalebar: 40 μm. Further observations of TTR localization within the cytoplasm of HCPECs. (B) Colocalization of α-Tubulin (red) and TTR (green); nuclei stained with DAPI (blue). Magnification: 60x, Scalebar: 40 μm. (C) Westernblot for the presence of CK18 and TTR (dimer). Lanes C1, C2, C3 –Uninfected HCPECs protein; Lanes I1, I2, I3 –Protein from HCPECs infected with *Bb* for 48 hours.

TTR within CK18 labeled cells; Fig 2B further illustrates the cytoplasmic localization of TTR by using α-Tubulin to highlight cellular boundaries. Total protein from control and infected groups was isolated to check for the presence of CK18 and TTR (Fig 2C). The effects of *Bb* infection on HCPEC proliferation, apoptosis, and necrosis was measured through IHC and colorimetric assays—no significant changes were detected between infected and control groups.

## Stimulation of type I/II interferon signaling pathway following B. burgdorferi infection

*B. burgdorferi* infection was performed with primary human choroid plexus epithelial cell cultures for 48 hours, and changes in the transcriptome of the human cells were compared to untreated controls (3 biological replicates per group). Supernatant from each replicate was collected and used for protein analysis, and RNA was then isolated. Following RNA isolation, Illumina libraries were made and sequenced on an Illumina HiSeq 4000 as outlined in the Methods section. Differential gene expression analysis was performed using the DESeq2 package in Rstudio. Following count normalization, an MA-plot (Fig 3A) was constructed–this illustrates genome-wide transcriptome expression of the 48-hour *Bb* treated group when compared to the control group. Dots in red represent significant differentially expressed genes

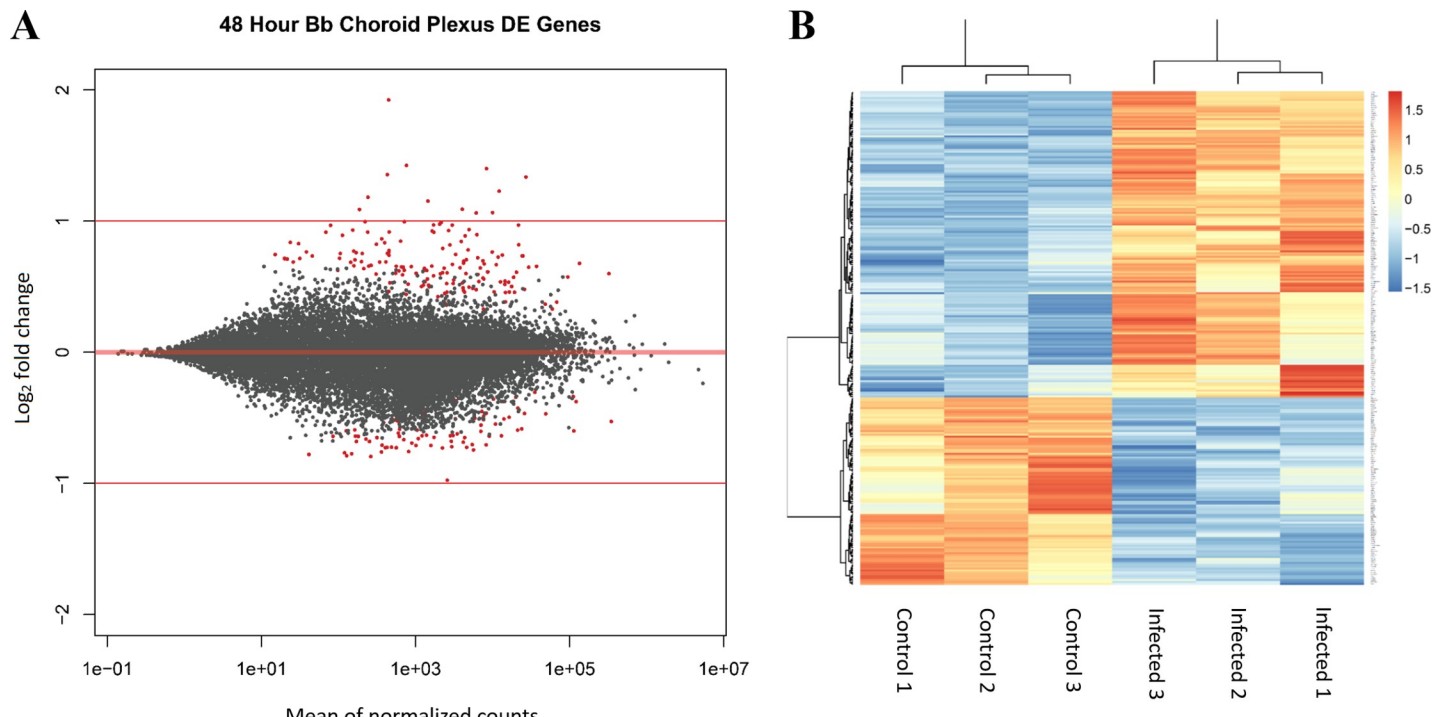

**Fig 3. RNA-seq was performed on HCPECs that were infected by *B. burgdorferi* for 48 hours and from uninfected controls.** (A) MA-plot representing gene expression patterns of infected group compared to control (n = 3). Red dots indicated significant differentially expressed genes (DEGs; FDR <0.05). A total of 258 genes were differentially expressed. (B) A heatmap of the 258 DEGs showing clustering patterns between each biological replicate.

(DEGs), plotted as a function of its log$_2$ fold change versus the mean of normalized counts. Using an adjusted p-value (false discovery rate) less than or equal to 0.05, and no fold change threshold, a total of 258 genes were shown to have significant differential expression (S1 File). Of these 258 DEGs, 160 were upregulated and 98 downregulated. A principal component analysis (PCA) plot was generated and shows clustering between treatment groups (S1 Fig). Hierarchical clustering of replicates based on treatment is further illustrated in the heatmap analysis (Fig 3B) of the 258 DEGs for each replicate.

In order to validate our RNA-seq findings, RT-qPCR was performed on a set of select genes (Fig 4A). The upregulated genes *cxcl3*, *cxcl6*, *ccl5*, *ifit1*, and *ifitm1* were found to be significantly upregulated by RT-qPCR when comparing 48-hour infected group to the untreated group. Although *cxcl5* and *irf7* transcripts were not significantly increased, an increasing trend was observed that correlates with the RNA-seq data. Additionally, the downregulated genes *cdh2*, *flt1*, and *anxa1* showed a non-significant downward trend that corresponds with our previous data. In order to determine if transcriptional changes observed in response to *Bb* resulted in protein production and secretion from the choroid plexus epithelial cells, we measured the cytokine levels in cell culture supernatants by ELISA (Fig 4B–4E). Following 48-hours post *B. burgdorferi* stimulation, Cxcl1, Cxcl2, Cxcl5, and Cxcl6 supernatant concentrations were assessed. Cxcl1, Cxcl2, and Cxcl5 were induced and secreted at significantly higher levels in the infected samples compared to untreated controls. Though Cxcl6 did not show significant elevation, a similar increase in protein levels was observed. These data agree with our RNA-seq findings.

Both subsets of upregulated and downregulated genes were analyzed for known interactions of their associated proteins using the STRING (Search Tool for the Retrieval of

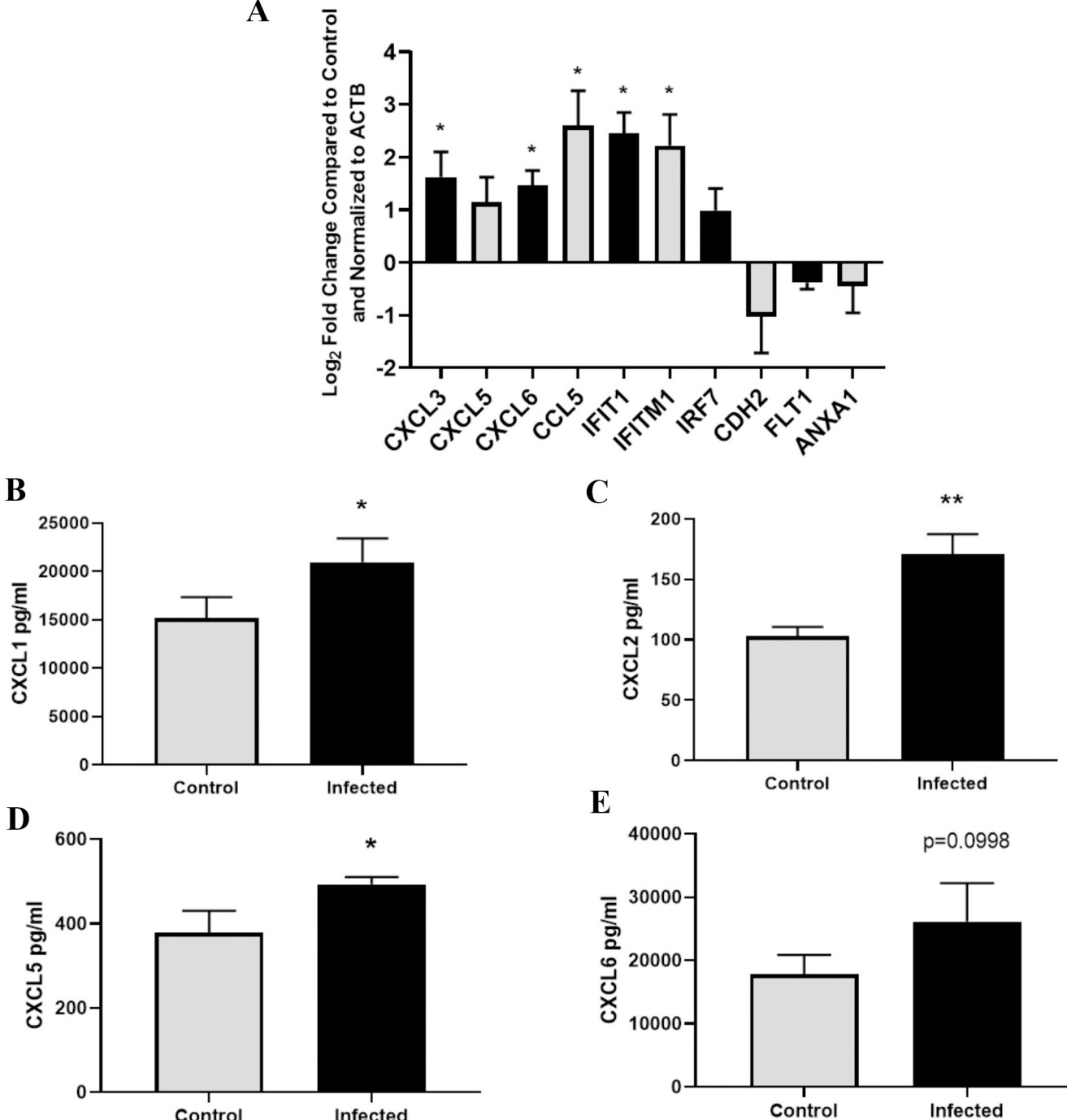

**Fig 4. Validation of RNA-seq gene expression data.** (A) Select differentially expressed genes following 48-hour *B. burgdorferi* treatment were validated by RT-qPCR. Primers specific to *cxcl3*, *cxcl5*, *cxcl6*, *ccl5*, *ifit1*, *ifitm1*, *irf7*, *cdh2*, *flt1*, *anxa1*, and *actb* were used for amplification. The relative gene expression between 48-hour *B. burgdorferi* treated and untreated control groups are expressed as Log2FC and normalized to the housekeeping gene ACTB. Significance was calculated by Student's t-test (* p<0.05, n = 3). (B-E) Supernatant from infected and untreated groups were collected and analyzed by ELISA. Concentrations of these secreted cytokines are shown as the mean and standard deviation. Significance was calculated by Student's t-test followed by Benjamini, Krieger, Yekutieli correction (* FDR < 0.05, n = 3).

Interacting Genes/Proteins - https://string-db.org/) database (Figs 5 and 6) [43]. STRING analysis provides insight into the protein-protein interactions of each gene–each node represents one of the DEGs; lines between nodes correspond to known interactions that were experimentally determined or curated from databases utilized by STRING. STRING also provides information on functional and pathway analysis through the use of Gene Ontology (GO) [45, 46] and KEGG [49]. Upregulated DEGs (Fig 5A) show two distinct clusters within the network. The first cluster, shown in the top left, is predominantly comprised of chemokines and cytokines–the red nodes denote genes within the chemokine-mediated signaling pathway (GO:0070098, FDR 8.23E-08). Additionally, the second highly clustered nodes situated within the center indicates genes associated with type I (blue) and type II (green) interferon pathways (GO:0060337, FDR 2.71E-23; GO:0034341, FDR 9.58E-15, respectively). The biological processes that were enriched predominantly involved inflammatory/immune response pathways and cell-cell communication (Fig 5B). A large subset of the upregulated genes was found to be interferon pathway genes (Table 1). Of note, the interferon-induced protein family of genes, *ifit1*, *ifit3*, *ifitm1*, *ifitm2*, *ifitm3*, and others, were shown to be significantly upregulated. Type I and II interferon related genes, such as *gbp2*, *ifit1*, and *oasl*, have been previously reported as being significantly elevated as a consequence of *B. burgdorferi* infection [52, 53]. Furthermore, major transcription factors that were observed to be upregulated including *irf7*, *stat1*, and *stat2*, have been shown to play important roles in regulating the *Bb*-induced interferon response [54–56]. Interferon related genes have traditionally been associated with viral, not bacterial, infections, and this is reflected in regard to the labeling of gene function and pathway analysis within these data. However, it is well documented that such genes are often observed to play key roles during bacterial infections, including Lyme disease [57–60]. Signaling Pathway Impact Analysis (SPIA) was performed to determine pathway enrichment based on an increase in gene enrichment and position of genes within a pathway. A total of 14 pathways were identified as being significantly enriched (Fig 7). The top activated pathways involve the activation of viral pathways, including Influenza A, Measles, and Herpes simplex infection. As stated before, these viral pathways involve the interferon related response as observed in our data. These data imply that infection with *B. burgdorferi* produces a significant immune response that encapsulates major interferon-signaling pathways within HCPECs.

## B. burgdorferi infection induces a chemokine profile in HCPECs conducive to the chemotaxis of immune cells

Many of the transcripts that were upregulated were categorized into pro-inflammatory cytokines and chemokines (Table 1)—these involved the C-X-C and C-C motif family of chemokines. *cxcl1*, *cxcl2*, *cxcl3*, *cxcl5*, and *cxcl6* showed elevated levels in response to *B. burgdorferi* infection. In addition to their role in modulating immune cell activation and inflammation, these chemokines provide a mechanism for the chemotaxis of immune cells. Predominantly, the C-X-C family possesses chemoattractant properties for leukocytes, such as neutrophils [61–63]. In fact, Cxcl1 mediates the recruitment of neutrophils and subsequent swelling in Lyme arthritis and carditis [64]. In contrast, the C-C family induces the migration of PBMCs, including lymphocytes and monocytes [65]. In our experiments, *ccl2*, *ccl5*, *ccl13*, and the receptor, *ccr7*, were found to be upregulated. This is corroborated in a study that previously showed *ccl2* (*mcp-1*) and *ccl5* (*rantes*) were elevated in human monocytes in response to *B. burgdorferi* [66]. Ccr7, a receptor for Ccl19 and Ccl21, is constitutively expressed in intestinal and gastric epithelial cells and has been shown to be upregulated in response to *Helicobacter pylori* infection [67, 68]. Biological process analysis performed by GO, via STRING, indicates cytokine and chemokine-mediated signaling pathways to be significantly enriched. Moreover, processes

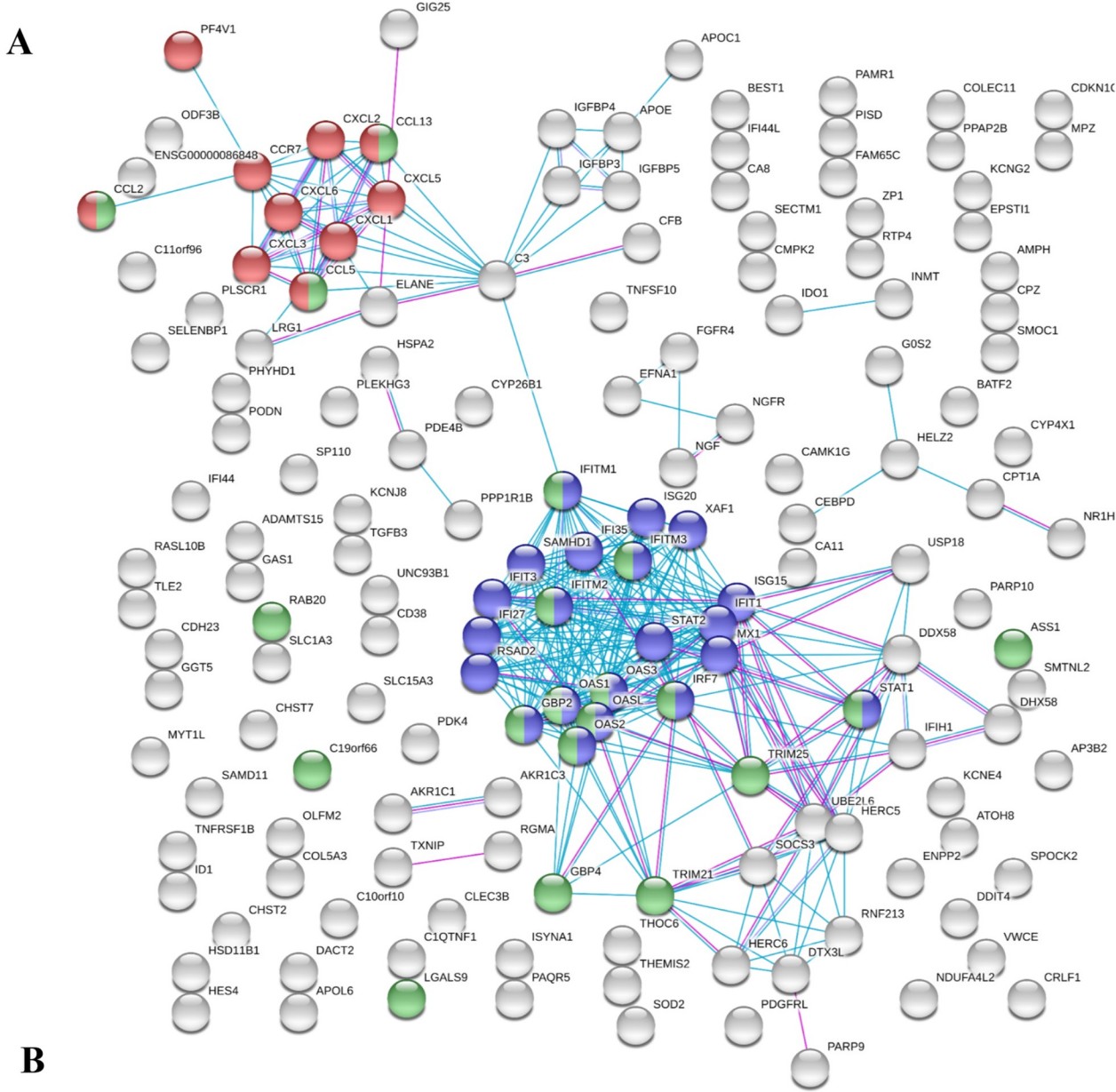

| GO Enrichment | FDR | GO Term |
|---|---|---|
| Response to type I interferon | 3.75E-24 | GO:0034340 |
| Type I interferon signaling pathway | 2.71E-23 | GO:0060337 |
| Innate immune response | 1.61E-21 | GO:0045087 |
| Cytokine-mediated signaling pathway | 4.29E-21 | GO:0019221 |
| Immune response | 4.67E-20 | GO:0006955 |
| Response to interferon-gamma | 9.58E-15 | GO:0034341 |
| Inflammatory response | 1.67E-10 | GO:0006954 |
| Chemokine-mediated signaling pathway | 8.23E-08 | GO:0070098 |
| Regulation of leukocyte chemotaxis | 2.31E-06 | GO:0002688 |
| Positive regulation of neutrophil chemotaxis | 7.71E-07 | GO:0090023 |

**Fig 5. Network and functional analysis of upregulated DEGs by STRING analysis.** (A) String network–Each node represents a DEG; lines between nodes indicate known protein-protein interactions. Nodes in blue and green correspond to type I or type II interferon pathways, respectively; Red corresponds to chemokine-mediated signaling pathway. (B) Table of select GO enrichments.

involving the regulation of leukocyte and neutrophil chemotaxis were shown to be enriched as well (Fig 5B). These observations are further strengthened by SPIA, where the pathways cytokine-cytokine receptor interaction and chemokine signaling pathways were found to be significantly enriched and activated (Fig 7). Further evidence that may imply the immune trafficking role of HCPECs comes from the secretion of these proteins at elevated levels within the culture media, as previously stated (Fig 3B–3E).

## B. burgdorferi effects on cellular components involved in cell-cell junctions and adhesion

Although HCPECs were grown in a non-barrier monolayer culture, it was found that a number of genes related to cellular junctions and adhesion were modestly downregulated (Table 2). The integrity of the BCSFB at choroid plexus epithelium is contingent on the presence of several tight and adherens junctions. Adherens junctions, found more basal than tight junctions (Fig 1), mainly involve cadherin proteins, for example E-cadherin, VE-cadherin, and N-cadherin [69]. The presence of CDH2(N-cadherin) has been observed on the basolateral side of the choroid plexus epithelium in mice [70]. Three genes within the cadherin superfamily were found to be downregulated–*cdh2*, *pcdh7*, and *pcdh10*. Likewise, genes that code for tight junction components showed lowered expression–*cldn14* and *magi1*. Genes within regulatory pathways that promote the formation of these junctions or other cellular adhesins were also found to have decreased expression–*mtss1*, *atp1b1*, and *frmd4a*. Network analysis showed minimal clustering of genes involved in cellular adhesion regarding protein-protein interactions (Fig 6A). Additionally, genes involved in the modulation of surrounding extracellular matrix and vasculature were observed to be downregulated, some of which shared overlapping function with cellular adhesion–*mmp1*, *flt1*, *vegfc*, and *serpine1*. GO enrichment indicated enriched cellular components that involve cell-cell junction and bicellular tight junctions, as well as angiogenesis and epithelium development processes. Pathway analysis indicates an inhibition of the focal adhesion pathway (Fig 7). The functional and structural impact of these downregulated genes in response to *B. burgdorferi* infection is yet to be determined in an animal model.

## Discussion

The neurological symptoms associated with Lyme disease are largely attributed to the dissemination of *Borrelia burgdorferi* into the CNS and the resulting host immune response. While previous studies have investigated the effects of this bacteria on endothelial models of the BBB, little is known about its impact on the epithelium of the choroid plexus which comprises the BCSFB. The CP epithelium is situated at a key interface that separates the blood from the CSF and has repeatedly been shown to play an important role in modulating the immune response between the periphery and CNS during infection. The importance of the composition of the CSF in regards to cytokines and infiltrating immune cells in Lyme disease patients has been previously reported [12, 23, 71–73]. However, in the context of Lyme disease, the choroid plexus has been greatly understudied, and given its role as the major producer of CSF, as well as its ability to regulate its composition, it constitutes a major gap in knowledge for the pathophysiology of the disease [74]. To the best of our knowledge, this is the first study to directly investigate the impact of *Borrelia burgdorferi sensu lato* on choroid plexus epithelium.

**A**

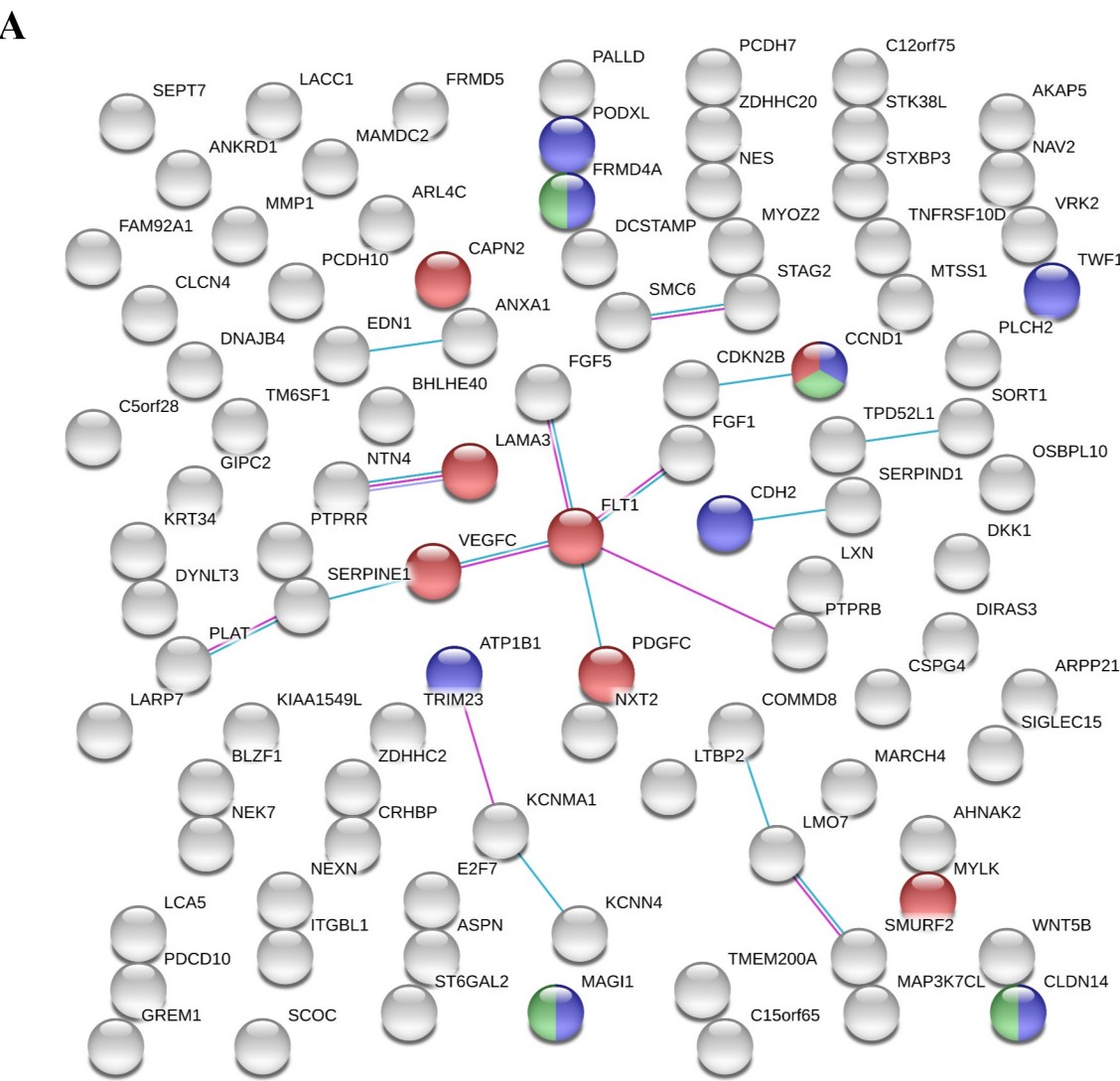

**B**

| GO Enrichment | FDR | Go Term |
|---|---|---|
| Regulation of cell migration | 0.00047 | GO:0030334 |
| Blood vessel morphogenesis | 0.0006 | GO:0048514 |
| Angiogenesis | 0.00082 | GO:0001525 |
| Epithelium development | 0.0127 | GO:0060429 |
| Cell-cell junction | 0.0244 | GO:0005911 |
| Bicellular tight junction | 0.0492 | GO:0005923 |

**Fig 6. Network and functional analysis of downregulated DEGs by STRING analysis.** (A) String network–Nodes in blue and green represent cell-cell junctions (GO:0005911) and bicellular tight junctions (GO:0005923), respectively. Nodes in red indicate genes within the focal adhesion pathway from KEGG pathway analysis (hsa04510). (B) Table of selected GO enrichments.

This study demonstrates a robust change in gene expression in HCPECs induced by *B. burgdorferi* infection. The most prevalent outcome was the upregulation in immune and inflammatory response genes that were primarily categorized within the chemokine/cytokine

**Table 1. Select inflammatory and immune response genes.**

| | Gene Symbol | Log$_2$ Fold Change | p-value (adjusted) | Function (Uniprot ID) |
|---|---|---|---|---|
| Interferon Related | OASL | 1.923812 | 7.63E-18 | Type I/II interferon signaling pathway–RNA binding (Q15646) |
| | IFITM1 | 1.336366 | 1.99E-13 | Type I/II interferon signaling pathway–Inhibits entry of virus (P13164) |
| | IFIT3 | 1.228324 | 6.81E-11 | Type I/II interferon signaling pathway–Inhibits viral processes (O14879) |
| | IFIT1 | 1.4001 | 1.51E-10 | Type I interferon signaling pathway–RNA binding (P09914) |
| | RSAD2 | 1.424712 | 7.87E-09 | Type I interferon signaling pathway–CD4+ T-cell activation (Q8WXG1) |
| | OAS2 | 1.064886 | 3.93E-06 | Type I/II interferon signaling pathway–Innate antiviral response (P29728) |
| | OAS1 | 1.061772 | 1.35E-05 | Type I/II interferon signaling pathway–Innate antiviral response (P00973) |
| | IFITM3 | 0.677765 | 2.45E-05 | Type I/II interferon signaling pathway–Inhibits entry of virus (Q01628) |
| | IFI27 | 0.865445 | 0.000102 | Type I interferon signaling pathway–Innate immune response (P40305) |
| | IFITM2 | 0.57169 | 0.000815 | Type I/II interferon signaling pathway–Inhibits entry of virus (Q01629) |
| | IRF7 | 0.764681 | 0.004436 | Type I/II interferon signaling pathway–Key transcriptional regulator (Q92985) |
| | OAS3 | 0.72591 | 0.008201 | Type I/II interferon signaling pathway–Innate antiviral response (Q9Y6K5) |
| | IFI35 | 0.759788 | 0.012529 | Type I interferon signaling pathway (P80217) |
| | GBP2 | 0.400131 | 0.049661 | Type I/II interferon signaling pathway–anti-pathogen activity (P32456) |
| | STAT1 | 0.734634 | 1.89E-05 | Signal transducer–Mediates interferon response (P42224) |
| | STAT2 | 0.45657 | 0.026725 | Signal transducer–Mediates interferon response (P52630) |
| Chemokine/cytokine Related | CCL5 | 1.355304 | 4.67E-09 | Chemotaxis–Monocytes, T-helper cells, eosinophils, neutrophils (P13501) |
| | CXCL2 | 0.927644 | 3.00E-07 | Chemotaxis–Leukocytes, neutrophils (P19875) |
| | CXCL1 | 0.734937 | 1.35E-05 | Chemotaxis–Neutrophils (P09341) |
| | CXCL6 | 0.654966 | 2.30E-05 | Chemotaxis–Neutrophils, leukocytes (P80162) |
| | CCL2 | 0.479371 | 0.002206 | Chemotaxis–Monocytes, basophils (P13500) |
| | CXCL3 | 0.837774 | 0.005424 | Chemotaxis–Leukocytes, neutrophils (P19876) |
| | ELANE | 0.819051 | 0.007334 | Modulates natural killer cells, monocytes, granulocytes, neutrophils (P08246) |
| | CXCL5 | 0.652521 | 0.009551 | Chemotaxis–Neutrophils, leukocytes (P42830) |
| | CCR7 | 0.659729 | 0.009835 | Chemokine receptor–mediates immune cell chemotaxis (P32248) |
| | CCL13 | 0.714727 | 0.032672 | Chemotaxis–Monocytes, lymphocytes, basophils, eosinophils (Q99616) |
| | C3 | 0.68928 | 0.037701 | Complement system (P01024) |

mediated pathways and type I and II interferon pathways. Consistent with our report, previous studies have observed similar results regarding the inflammatory and immune response within monocytes, macrophages, and dendritic cells, showing an increase in cytokines such as *cxcl1*, *cxcl2*, *ccl2*, and *ccl5* [75, 76]. Likewise, interferon-stimulated genes within a murine model were reported to be upregulated, involving the transcripts *Ifit1*, *Ifit3*, and *Irf7* [55]. Irf7 has been shown to be a master regulator of interferon stimulated genes, and in conjunction with the upregulation of *ddx58* (*rig-I*), *ifih1* (*mda4*), and *trim25*, may provide insight into the activation of the interferon pathway being observed. Additionally, the induction of inflammatory cytokines including type I and type III interferons were reported when human PBMCs were infected with *B. burgdorferi* [77]. Furthermore, when characterizing the immunophenotypes of infiltrating immune cells and cytokines within the erythema migrans (EM) lesions of patients, T cells, monocytes, macrophages, and dendritic cells were found to be enriched in addition to inflammatory cytokines [78].

While this study shows overlapping features common in other cell types or animal models infected with *Bb*, the importance of the choroid plexus's role in immune cell trafficking is further highlighted in other models of infection and disease. The concept of the immunosurvellience activity within the choroid plexus is not new, and the abundance of immune cells found within the choroid plexus and subsequent transmigration following infection or insult

**A**

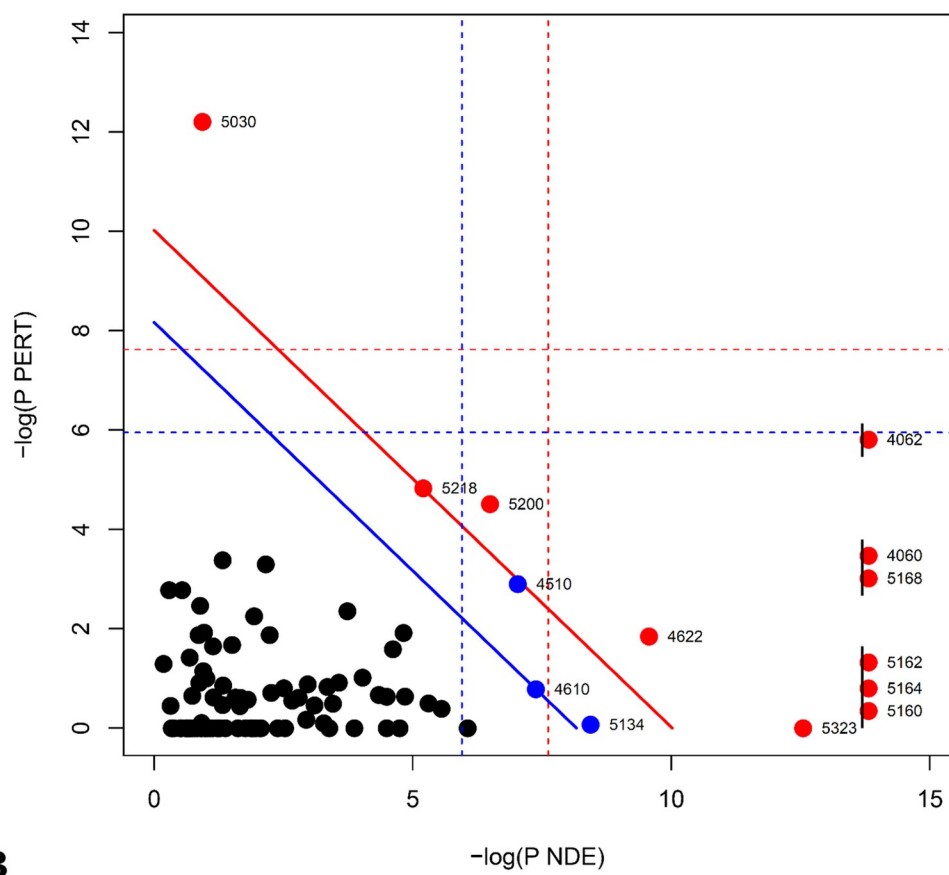

**B**

| Pathway ID | Pathway | pGFdr | Status |
|---|---|---|---|
| 5164 | Influenza A | 2.33E-10 | Activated |
| 4060 | Cytokine-cytokine receptor interaction | 7.84E-09 | Activated |
| 5162 | Measles | 1.75E-07 | Activated |
| 4062 | Chemokine signaling pathway | 6.71E-07 | Activated |
| 5168 | Herpes simplex infection | 7.51E-07 | Activated |
| 5160 | Hepatitis C | 1.76E-05 | Inhibited |
| 5030 | Cocaine addiction | 0.000405 | Inhibited |
| 5323 | Rheumatoid arthritis | 0.000612 | Inhibited |
| 4622 | RIG-I-like receptor signaling pathway | 0.00155 | Activated |
| 5200 | Pathways in cancer | 0.002034 | Inhibited |
| 5218 | Melanoma | 0.004492 | Inhibited |
| 4510 | Focal adhesion | 0.004505 | Inhibited |
| 5134 | Legionellosis | 0.014968 | Inhibited |
| 4610 | Complement and coagulation cascades | 0.018952 | Activated |

**Fig 7. Signaling pathway impact analysis.** SPIA of all DEGs based on pathway gene enrichment (pNDE) and pathway perturbations (pPERT) that take into account gene placement and topology within the pathway. Both pNDE and pPERT are used to determine global significance, pG. (A) SPIA two-way evidence plot. Each dot represents a pathway that contains at least one DEG. The impact analysis plots each pathway based on pNDE and pPERT. Pathways above the solid blue line are significant following FDR correction (pGFDR < 0.05). Pathways above the solid red line are significant following Bonferroni correction (pGFWER < 0.05). (B) A table of all significant pathways (PGFDR <0.05) and their respective status is shown.

**Table 2. Select genes involved in cell-cell junctions, tight junctions, and adherens junctions.**

| | Gene Symbol | Log$_2$ Fold Change | p-value (adjusted) | Function (Uniprot ID) |
|---|---|---|---|---|
| Functional component | CLDN14 | -0.77849 | 0.014354 | Tight junction protein; Cell adhesion (O95500) |
| | PCDH10 | -0.70536 | 0.000243 | Protocadherin; cell-cell adhesion (Q9P2E7) |
| | CDH2 | -0.65003 | 3.70E-07 | Adherens junction protein; Cell adhesion (P19022) |
| | MAGI1 | -0.42719 | 0.03935 | Scaffolding/Tight Junction Protein; Cell adhesion (Q96QZ7) |
| | PCDH7 | -0.39693 | 0.011885 | Protocadherin; cell-cell adhesion (O60245) |
| Regulatory component | PODXL | -0.97617 | 6.91E-07 | Positive/negative regulation of cell adhesion (O00592) |
| | TWF1 | -0.67896 | 0.038453 | Actin binding; Cadherin binding; Focal adhesion (Q12792) |
| | NEXN | -0.63395 | 0.015809 | Actin binding protein; Cell adhesion (Q0ZGT2) |
| | MTSS1 | -0.53155 | 0.000231 | Actin binding protein; Positive regulation of cell-cell junctions, adhesion (O43312) |
| | FLT1 | -0.50364 | 0.01255 | VEGF receptor; Endothelial proliferation, survival, cell adhesion (P17948) |
| | ATP1B1 | -0.47766 | 0.034878 | ATPase non-catalytic beta subunit; Cell adhesion; Epithelial cell polarity (PO5026) |
| | MYLK | -0.43527 | 0.014211 | Regulates tight junctions; Regulates epithelial cell survival, wound healing (Q15746) |
| | CCND1 | -0.37679 | 0.008102 | Regulates cell cycle; Interactions with tight junction proteins (P24385) |
| | FRMD4A | -0.3571 | 0.017093 | Scaffolding Protein–Regulates epithelial cell polarity, adherens junctions (Q9P2Q2) |
| | CAPN2 | -0.33762 | 0.047708 | Protease; Negative regulation of junction and adhesive pathways (P17655) |
| | PALLD | -0.30542 | 0.041217 | Scaffolding/Cytoskeletal protein; Cell adhesion (Q8WX93) |

has been widely reported [27, 30, 32, 79, 80]. Functional and pathway analysis indicates that many of these genes are involved in chemotaxis of immune cells [61, 62, 81–84]. In fact, *Cxcl1* (previously known as *Kc*) and *Ccl2* (previously known as *Mcp-1*) have been shown to mediate the recruitment of neutrophils into the joints of mice infected with *B. burgdorferi*, and are required for the development of Lyme arthritis [85]. The upregulation of *cxcl1*, *cxcl2*, *cxcl3*, *ccl2*, and *ccl5*, potent chemoattractants for immune cells including neutrophils, monocytes, and T cells, among others, have also been consistently reported to be elevated in other bacterial infections of the CP. In a barrier model of the choroid plexus involving infection with *Neisseria meningitidis*, in addition to an increase in these chemoattractants, the recruitment and subsequent transmigration of polymorphonuclear neutrophils and monocytes was observed [86, 87]. Similar results are also seen in response to *Streptococcus suis*, a gram-positive bacterium that can be transmitted to humans from pigs, leading to symptoms such as meningitis [88]. In a BCSFB model using human choroid plexus papilloma cells, a viral infection with Echovirus 30 showed an enhanced secretion of Cxcl1, Cxcl2, Cxcl3, and Ccl5 [89]. Indeed, when investigating the composition of the CSF from individuals with *B. burgdorferi* induced meningoradiculitis, an increase in inflammatory cytokines and a large number of B cells and plasma cells are observed [90]. Collectively, the choroid plexus has been shown to contribute significantly to the pathogenesis of many diseases and in regards to Lyme disease, our data implies that the choroid plexus may play an important role in the abundance of immune cell invasion of the CSF and the exacerbation of CNS inflammation that is seen in patients [91, 92].

In our current model, our transcriptome analysis indicates a downregulation of key tight junction and adherens junction genes, as well as regulatory cell adhesion genes. *cldn14*, a tight junction protein associated with the choroid plexus, as well as *cdh2*, an adherens junction protein, were found to be downregulated [93, 94]. The injection of LPS in mice to stimulate a peripheral inflammatory response showed a similar gene expression pattern, where the majority of upregulated genes within the choroid plexus involved immune-mediated pathways, while downregulated genes participated in barrier function, including claudins and protocadherins [95]. Though the functions of protocadherins are still being fully elucidated, they have been found to play a key role in cellular adhesion and barrier integrity [96, 97]. In addition, the

sodium/potassium transporter beta subunit, *atp1b1*, was found to be downregulated, yet, seems to be an unlikely participant in the formation of cell-cell junctions; however, it has been shown to play an integral part in cell adhesion and in both the formation and maintenance of tight junctions in epithelial cells [98–101]. The downregulation of a number of these components, as well as scaffolding and other regulatory genes such as *magi1* and *mtss1*, would indicate a potential dysregulation of the choroid plexus barrier. This may lead to the possibility of immune cell invasion as well as an entry site for *Borrelia burgdorferi* into the CSF. In fact, the choroid plexus has already been implicated as a possible site of entry for both *N. meningitidis* and *S. suis*, specifically from the basolateral side [102]. By using two *in vitro* barrier models constructed by human brain microvascular endothelial cells (BMEC) and umbilical vein endothelial cells (HUVEC), *Bb* was found to differentially transverse these barrier systems [103–105]. While *B. burgdorferi*was capable of crossing the HUVEC monolayer, the traversal of the bacteria across the BMEC barrier required the addition of plasminogen and was found to induce the expression of plasminogen activators, receptors, and matrix metalloproteinases–supporting the concept that the bacteria is able to utilize the fibrinolytic system which may promote its dissemination through the degradation of the extracellular matrix and cell-to-cell junctions [103–105]. However, in our experiments, we found conflicting results–tissue plasminogen activator, tPA (*plat*), as well as its inhibitor, *serpine1*, were both found to be down regulated. Furthermore, the metalloproteinases, *mmp1*, and *adamts15*, were found to be downregulated and upregulated, respectively. Though our system does not represent a barrier model, the observed outcome lends credibility for our future studies involving *B. burgdorferi* infection in an *in vivo* model to explore the impact of systemic infection on the BCSFB.

## Conclusion

Following infection of HCPECs with *Borrelia burgdorferi*, we identified a gene expression pattern that is marked by a robust increase in immune and inflammatory genes within the cytokine/chemokine pathways and type I and II interferon pathways. Protein analysis showed an enhanced secretion of these inflammatory and chemotactic cytokines. Additionally, the downregulation of genes involved in cell-cell adhesion, adherens junctions, and tight junctions was observed. Overall, our data indicates that the choroid plexus, like in many other infectious diseases, may play a key role in the pathogenesis of Lyme neuroborreliosis through the induction of inflammatory factors, the promotion of immune cell migration, and potentially through the dysregulation of the BCSFB. Our future studies will aim to elucidate the impact of *B. burgdorferi* infection on the BCSFB integrity within an *in vivo* model. By understanding how the inflammatory and immune response is modulated within the CNS, as well as mechanisms by which *Borrelia burgdorferi* is able to traverse into the CNS, new treatments for Lyme disease can be developed.

## Supporting information

**S1 File. Table of all 256 DEGs and supporting data.**
(CSV)

**S2 File.**
(XLSX)

**S1 Fig. Principle component analysis.**
(PNG)

**S1 Raw images.**
(PDF)

## Acknowledgments

We thank the University of North Dakota Genomics Core for quality control and technical support.

## Author Contributions

**Conceptualization:** John A. Watt.

**Investigation:** Derick Thompson, Jordyn Sorenson, Jacob Greenmyer, Catherine A. Brissette, John A. Watt.

**Supervision:** Catherine A. Brissette, John A. Watt.

**Writing – original draft:** Derick Thompson.

**Writing – review & editing:** Catherine A. Brissette.

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
