## [Decision Letter · Decision Letter 0]

19 Feb 2020

PONE-D-20-02265

The Lyme disease bacterium, *Borrelia burgdorferi*, stimulates an inflammatory response in human choroid plexus epithelial cells

PLOS ONE

Dear Dr. Watt,

Thank you for submitting your manuscript to PLOS ONE. After careful consideration, we feel that it has merit but does not fully meet PLOS ONE’s publication criteria as it currently stands. Therefore, we invite you to submit a revised version of the manuscript that addresses the points raised during the review process.  The main issues focus on the number of experiments/samples performed for each figure and whether the appropriate statistical analyses were performed on these data.  

We would appreciate receiving your revised manuscript in the next few months. To enhance the reproducibility of your results, we recommend that if applicable you deposit your laboratory protocols in protocols.io, where a protocol can be assigned its own identifier (DOI) such that it can be cited independently in the future. For instructions see: http://journals.plos.org/plosone/s/submission-guidelines#loc-laboratory-protocols

We look forward to receiving your revised manuscript.

Kind regards,

R. Mark Wooten, Ph.D.

Academic Editor

PLOS ONE

Journal Requirements:

1. We note that you have stated that you will provide repository information for your data at acceptance. Should your manuscript be accepted for publication, we will hold it until you provide the relevant accession numbers or DOIs necessary to access your data. If you wish to make changes to your Data Availability statement, please describe these changes in your cover letter and we will update your Data Availability statement to reflect the information you provide.

Reviewers' comments:

Reviewer's Responses to Questions

**Comments to the Author**

1. Is the manuscript technically sound, and do the data support the conclusions?

Reviewer #1: No

Reviewer #2: Yes

2. Has the statistical analysis been performed appropriately and rigorously? 

Reviewer #1: No

Reviewer #2: Yes

3. Have the authors made all data underlying the findings in their manuscript fully available?

Reviewer #1: Yes

Reviewer #2: No

4. Is the manuscript presented in an intelligible fashion and written in standard English?

Reviewer #1: Yes

Reviewer #2: Yes

5. Review Comments to the Author

Reviewer #1: The manuscript by Thompson et al assesses inflammatory responses of human choroid plexus epithelial cells in response to stimulation with the Lyme disease spirochete Borrelia burgdorferi. The major weakness with this manuscript is that the authors did not establish a fold-change cutoff to their RNA-seq data. This makes the analysis highly susceptible to Type 1 error, particularly where only 3 biological replicates were used for each condition. The authors claimed that several hundred genes were differentially expressed, but the vast majority of these had a very modest fold-change in expression, casting doubt on the biological significance of these results. In addition, qRT-PCR and cytokine data show minor changes in expression/protein levels, and the manuscript suffers from over-stating these modest changes that may or may not be biologically relevant. In order to be appropriate for publication, a much more critical analysis of the data will be required. The manuscript would also be greatly enhanced by increasing the number of samples for each group. Figure 3 shows marked variability between samples, which suggests much of the data are simply statistical noise.

Several minor concerns are noted below:

1. Please use the correct the mouse nomenclature for genes, transcripts, and proteins as appropriate

2. Brain anatomy needs to be labeled in Fig 1 (particularly the choroid plexus)

3. Methods needs a statistical analysis section clearly describing the types of statistical analyses performed for each experiment

4. qRT-PCR data lack any post-hoc analysis

5. There are numerous abbreviations, many of which are used without spelling them out the first time they are introduced

6. The manuscript would be easier to read if some of the abbreviations were eliminated.

Reviewer #2: The report entitled "the Lyme disease bacterium, Borrelia burgdorferi, stimulates an inflammatory response in human choroid plexus epithelial cells" describes a transcriptomic analysis of this key nervous system cell type in response to stimulation with Bb. Overall, the paper is well-organized and written. The hypothesis is clearly stated and in general, the conclusions are supported by the data. A few minor concerns/changes are noted.

-Perhaps it was missed, but the raw data should be uploaded into a public data base.

-p10, line 174-175 "samples were pooled.." How many?

-line 167 samples were "run"

-p11 line 209 spell out actb

-in Fig. 4, what is "n"? What do the black versus grey bars represent?

-p19 line 372 genes with cellular junctions were "modestly" downregulated.

-p23 line 438 reference needed. What are CP papilloma cells?

-(throughout": our data "suggest"

-p24, line 464 reference needed

6. PLOS authors have the option to publish the peer review history of their article (what does this mean?). If published, this will include your full peer review and any attached files.

Reviewer #1: No

Reviewer #2: No

---

## [Author Response · Author response to Decision Letter 0]

11 May 2020

Please find our response to the reviewers below, indicated by bullet points. 

Reviewer #1: The manuscript by Thompson et al assesses inflammatory responses of human choroid plexus epithelial cells in response to stimulation with the Lyme disease spirochete Borrelia burgdorferi. The major weakness with this manuscript is that the authors did not establish a fold-change cutoff to their RNA-seq data. This makes the analysis highly susceptible to Type 1 error, particularly where only 3 biological replicates were used for each condition. The authors claimed that several hundred genes were differentially expressed, but the vast majority of these had a very modest fold-change in expression, casting doubt on the biological significance of these results. In addition, qRT-PCR and cytokine data show minor changes in expression/protein levels, and the manuscript suffers from over-stating these modest changes that may or may not be biologically relevant. In order to be appropriate for publication, a much more critical analysis of the data will be required. The manuscript would also be greatly enhanced by increasing the number of samples for each group. Figure 3 shows marked variability between samples, which suggests much of the data are simply statistical noise.

• We appreciate the thorough and careful input provided by Reviewer #1, especially regarding statistical and biological significance of our experiments. 

• Regarding not establishing a fold-change cutoff: It is our understanding that establishing a fold-change cutoff is typically an arbitrary value and is usually used as a tool to reduce the number of differentially expressed genes to a manageable number (and selecting only the largest FC genes) for downstream analysis – which we felt was not necessary for our gene list (258 DEG’s). Therefore, selecting a fold-change cutoff would not remove any risk of a type 1 error, and would also introduce bias within our data. Additionally, a risk of a type 1 error is not directly tied to the magnitude of the fold change but more precisely to the raw read counts. In other words, it is more likely that a low transcript gene (i.e. 10 reads total) shows, by random chance, an increase of 10 reads within our treated group (10 -> 20 = log2FC of 1) than a high transcript read (i.e. 10,000 reads) showing the same log2FC of 1 (i.e. 10,000 -> 20,000). An example within in our data set would be PAMR1 – this gene shows a low log2FC of 0.383 but a very high read count (control = 58469, treated = 77514.14. Statistically and biologically speaking, we feel that our results for low log2FC genes are not inherently at any greater risk of a type 1 error than other genes with a higher log2FC. However, we agree with the reviewer that type 1 errors are always a concern in these datasets, especially if a gene shows a low read count. In fact, our methods for data and statistical analysis took this into consideration and aimed to reduce these risks – discussed below. 

• The above example regarding low transcript reads and type 1 errors is typically a concern in all transcriptome related experiments. However, many statistical packages, such as DESeq2 which was used in our analysis (Citation #42 in manuscript), corrects for these issues in a couple of different ways. Variability between replicates for each gene needs to be accurately modeled to test for differential expression. As quoted below from DESeq2 citation, studies with large sample sizes typically do not suffer from this short-coming, while smaller studies may show higher variability for each gene.

o “Accurate estimation of the dispersion parameter α i is critical for the statistical inference of differential expression. For studies with large sample sizes this is usually not a problem. For controlled experiments, however, sample sizes tend to be smaller (experimental designs with as little as two or three replicates are common and reasonable), resulting in highly variable dispersion estimates for each gene. If used directly, these noisy estimates would compromise the accuracy of differential expression testing.”1 – emphasis mine. 

o To correct for these shortcomings in studies such as ours, for any specific gene, the expected variability for that gene is modeled based on other genes within the entire genome that share a similar average expression strength. The underlying assumption provided by the authors of DESeq2 state that genes of similar average expression strength will have similar dispersion. If we go back to the previous example of the low transcript gene with a read count of 10, DESeq2 will find all genes with a similar read count, determine the overall variability of these genes, and apply that model to this low transcript gene. If all low transcript genes showed only a variability of 10 +/- 1, statistically, our gene with a low transcript read changing from 10 -> 20 would inherently have a lower risk of a type 1 error and the observed change may in fact be real. Based on this modeling, our gene would be corrected to have a “true” value change of 19 instead of the observed 20 – this is called “shrinkage estimation of dispersion” by the authors of DESeq2. The authors use an empirical Bayes approach which bases this shrinkage on two factors 1) the above-mentioned method of “true” dispersion, and 2) the degrees of freedom of the experiment – as we increase sample size, this shrinkage is reduced. Through this modeling, we can therefore correct and reduce the risk of a type 1 error for differentially expressed genes that show low transcripts as well as taking into account our sample size. 

o Furthermore, DESeq2 applies a post hoc statistical test, Benjamini-Hochberg adjustment, to reduce the likelihood of a type 1 error. All significant differentially expressed genes within the RNA-seq data use this adjusted p-value (padj) to determine significance – also called false discovery rate (FDR). A padj/FDR < 0.05 was considered as significant. 

o Based on the above explanation, we believe our statistical analysis of our RNA-seq data is quite conservative in its approach and is in-line with current standards. 

• The reviewer next brings into question the biological significance of the results. Again, we want to thank the reviewer for their attention to detail in tying together statistical and biological significance. We agree with the reviewer that a low log2FC, even if statistically significant, may not have any biological relevance. However, we feel that biological relevance should not be limited to just the fold change and therefore sought to determine functional and pathway analysis to determine biological significance. A single low expressed cytokine would not prompt an immune response typically seen in Lyme disease patients, but based on our downstream analysis (STRING clustering, Gene ontology for functional analysis, and pathway analysis via SPIA), we observed that a large percentage of upregulated genes fall into inflammatory and immune response groups (Fig 5). By showing that a large number of interconnected inflammatory/immune related genes are upregulated, we feel these pathways demonstrate biological significance. Additionally, as cited in the discussion section, this study shows similar results to other studies of Lyme disease in different animal/cell models. Similar results are also seen within other models of infection of the choroid plexus. 

o Regarding qRT-PCR and ELISA data: Both experiments were performed as a validation test of our RNA-seq data, as well as to determine if the observed transcriptome data translated to the protein level. The modest changes seen within both experiments mirror the changes seen in our RNA-seq and we view this as successful validation. 

o Because of these reasons, we believe biological relevance has been established. However, we agree with the reviewer that, regarding the downregulated genes, biological relevance may not be as robust. We were quite excited to observe changes in cell-cell adhesion genes and, as discussed in the manuscript, aim to bring these experiments into an animal model to further establish biological relevance in structural changes of the choroid plexus. Due to the limitations of this project, we had aimed to down-play the biological significance of these downregulated changes (without completely ignoring them) by placing the inflammatory and immune response observations as the core of the manuscript. 

• To overcome many of the stated shortcomings of the manuscript, the reviewer suggests increasing the number of samples. As briefly mentioned, as part of the modeling system of DESeq2, increasing the number of samples would in fact reduce variability of our system. While this would increase statistical power, it would not change the observed magnitude in log2FC. In fact, if we were to increase replicates, we would see a greater increase in the number of low log2FC genes as statistical power increased. 

o If additional replicates were to be added at this point, our data would suffer from batch effect and increase variability. Even during sequencing, flow cell variability as well as inter-lane variability would also impact the results. Our methods for this manuscript took this into consideration – all samples were pooled prior to shipment and therefore all samples were sequenced across two lanes in order to reduce variability. 

o Regarding the reviewer’s comment of Fig. 3: I am assuming the reviewer is addressing Fig 3B. The intended purpose of Fig 3B was to illustrate variability across replicates as well as across treatment groups. So, while there is some variability across replicates, there is much greater variability across treatment groups. This is evident based on the coloring scheme used, as well as the correlation clustering indicated at the top of the heatmap – showing that control replicates cluster together and Infected groups cluster together. 

Several minor concerns are noted below:

1. Please use the correct the mouse nomenclature for genes, transcripts, and proteins as appropriate

• Corrected nomenclature for mice were added in the following sections:

o Page 21, line 398

o Page 22, line 434

o Page 23, lines 449-450

2. Brain anatomy needs to be labeled in Fig 1 (particularly the choroid plexus)

• Figure 1 has been updated to reflect brain anatomy labels – Choroid Plexus and Ventricles

3. Methods needs a statistical analysis section clearly describing the types of statistical analyses performed for each experiment

• We appreciated the reviewer’s comment for the need of a statistical analysis section. Statistical analysis for each experiment was stated in its relevant section, however, as stated, it would add clarity to the methods to reorganize this into its own section. A statistical analysis section has been added and can be found as the last section of the Methods. The statistical analysis section explains the methods used for RNA-seq, qRT-PCR, and ELISA. 

4. qRT-PCR data lack any post-hoc analysis

• We have taken the reviewer’s advice and performed post-hoc analysis for multiple t-test comparisons. Specifically, we applied the Benjamini-Krieger-Yekutieli correction for determining the False Discovery Rate (FDR, p-adjusted)2. This is an improved procedure of their previous and standardized method, the Bejamini-Hochberg. Additionally, this updated procedure was the recommended correction provided by our statistical software (Graphpad PRISM) for multiple t-test comparisons. The outcome of this correction provided us with more conservative adjusted p-values and is reflected in the updated Figure 4. Significance was set to FDR < 0.05. 

5. There are numerous abbreviations, many of which are used without spelling them out the first time they are introduced

• We were not able to identify instances within the manuscript that used an abbreviation without spelling it out the first time it was introduced. However, there were two instances in which words were abbreviated and spelled out for the first time in the abstract as opposed to the main body of the paper. We reintroduced two abbreviates within the manuscript body:

o Page 5, line 62 – “cerebral spinal fluid (CSF)”

o Page 4, line 49 – “central nervous system (CNS)”

6. The manuscript would be easier to read if some of the abbreviations were eliminated.

• We appreciate the reviewer’s attention to readability and have tried to minimize unnecessary abbreviations. 

• Many abbreviations have been removed or changed, mainly involving “Bb” and “CP”, to read easier.

o Some instances of Bb were changed to either “B. burgdorferi” or “Borrelia burgdorferi”

o Some instances of CP were changed to “choroid plexus”.

Reviewer #2: The report entitled "the Lyme disease bacterium, Borrelia burgdorferi, stimulates an inflammatory response in human choroid plexus epithelial cells" describes a transcriptomic analysis of this key nervous system cell type in response to stimulation with Bb. Overall, the paper is well-organized and written. The hypothesis is clearly stated and in general, the conclusions are supported by the data. A few minor concerns/changes are noted.

-Perhaps it was missed, but the raw data should be uploaded into a public data base.

• The raw data will be uploaded to the GEO repository upon notice of acceptance. The accession number will be provided at that time. 

-p10, line 174-175 "samples were pooled.." How many?

• For RNA sequencing, there were 6 samples total – 3 Control (non-infected), and 3 Infected with Bb. The number of samples were briefly stated on Page 13, Line 249 of the original document; however, after review, the number of samples could be more clearly defined. To clarify the number of samples used, the following was added within the Methods section: 

o Page 8, Line 141-143: “Two groups were used for these experiments – Control, non-infected (N=3) and Infected, 48 hours (N=3); a total of 6 samples.”

o Page 10, Line 180-181, underlined the edited portion: “All samples, 3 control and 3 infected, were then pooled and sent to Novogene….”

-line 167 samples were "run"

• Corrected. “ran” to “run”

-p11 line 209 spell out actb

• As the sentence referred directly to the gene “…normalized to the housekeeping gene actb”, “actb” was kept but “β-actin” was added directly after in parentheses to clarify – “…normalized to the housekeeping gene actb (β-actin)”

-in Fig. 4, what is "n"? What do the black versus grey bars represent?

• The “n” for figure 4A is found in the figure legend – “n=3”, Page 15, Line 291. For Figure 4B-E, “n=3” was added, Page 16, Line 312.

• The black and grey bars do not represent anything and were simply added to visually enhance the figure. All bars are labeled accordingly at the bottom of each graph. 

-p19 line 372 genes with cellular junctions were "modestly" downregulated.

• “Modestly” was added to more accurately communicate the magnitude of change that was observed. 

-p23 line 438 reference needed. What are CP papilloma cells?

• Reference added, #88; all subsequent reference #’s updated

-(throughout": our data "suggest"

• We assumed the reviewer meant that the word “suggest” was used to often. We changed this word in a few instances to synonyms to make the manuscript more readable – “imply”, “indicate”, etc. 

-p24, line 464 reference needed

• Originally, references 103-105 refer to two sentences and was placed at the end of the second sentence. These references have been added to the end of both sentences.

References

1. Love, M. I., Huber, W. & Anders, S. Moderated estimation of fold change and dispersion for RNA-seq data with DESeq2. Genome Biol. 15, 550 (2014).

2. Benjamini, Y., Krieger, A. M. & Yekutieli, D. Adaptive linear step-up procedures that control the false discovery rate. Biometrika 93, 491–507 (2006).

---

## [Decision Letter · Decision Letter 1]

8 Jun 2020

The Lyme disease bacterium, Borrelia burgdorferi, stimulates an inflammatory response in human choroid plexus epithelial cells

PONE-D-20-02265R1

Dear Dr. Watt,

We’re pleased to inform you that your manuscript has been judged scientifically suitable for publication and will be formally accepted for publication once it meets all outstanding technical requirements.

Kind regards,

R. Mark Wooten, Ph.D.

Academic Editor

PLOS ONE

Additional Editor Comments (optional):

Reviewers' comments:

Reviewer's Responses to Questions

**Comments to the Author**

1. If the authors have adequately addressed your comments raised in a previous round of review and you feel that this manuscript is now acceptable for publication, you may indicate that here to bypass the “Comments to the Author” section, enter your conflict of interest statement in the “Confidential to Editor” section, and submit your "Accept" recommendation.

Reviewer #1: All comments have been addressed

Reviewer #2: All comments have been addressed

2. Is the manuscript technically sound, and do the data support the conclusions?

Reviewer #1: Yes

Reviewer #2: Yes

3. Has the statistical analysis been performed appropriately and rigorously? 

Reviewer #1: Yes

Reviewer #2: Yes

4. Have the authors made all data underlying the findings in their manuscript fully available?

Reviewer #1: Yes

Reviewer #2: Yes

5. Is the manuscript presented in an intelligible fashion and written in standard English?

Reviewer #1: Yes

Reviewer #2: Yes

6. Review Comments to the Author

Reviewer #1: This reviewer feels that while most of the concerns were adequately addressed, the response to the major issue of the number of biological replicates, while thorough, did not address the central concern of biological relevance. However, concerns brought up by the authors regarding batch effects of nexgen seq are legitimate. One possibility to overcome both issues would be to conduct a second independent stimulation experiment using more biological replicates and just use qRT-PCR and ELISA to test a limited number of target genes and cytokines. This is given as a suggestion only. Under the current circumstances related to the pandemic, one may well understand why conducting additional experiments at this time may be difficult or impossible. The data and findings, while imperfect, are important for the field and should be published.

Reviewer #2: (No Response)

7. PLOS authors have the option to publish the peer review history of their article (what does this mean?). If published, this will include your full peer review and any attached files.

Reviewer #1: No

Reviewer #2: No

---

## [Editor Report · Acceptance letter]

29 Jun 2020

PONE-D-20-02265R1 

The Lyme disease bacterium, Borrelia burgdorferi, stimulates an inflammatory response in human choroid plexus epithelial cells 

Dear Dr. Watt:

I'm pleased to inform you that your manuscript has been deemed suitable for publication in PLOS ONE. Congratulations! Your manuscript is now with our production department. 

Kind regards, 

on behalf of

Dr. R. Mark Wooten 

Academic Editor

PLOS ONE